# Application of Composite Materials in an Upgraded Engine Low-Pressure Compressor for a Regional Passenger Aircraft

**Yury Ravikovich, Alexander Arkhipov, Alexander Shakhov and Timur Erofeev ***

Moscow Aviation Institute, National Research University, 125993 Moscow, Russia; yurav2@yandex.ru (Y.R.); arkhipov.48@list.ru (A.A.); shakhov_alexander@mail.ru (A.S.)
**\*** Correspondence: erofeev_mai@mail.ru

**Abstract:** Computational and experimental studies have been carried out to evaluate the robustness and durability of components produced of polymer composite materials (PCM), as a part of the modernization of the low-pressure compressor (LPC) of the engine for the regional aircraft. For a preliminary assessment of the static and dynamic strength of the parts, a series of three-dimensional finite element calculations and tests of laboratory specimens, structural elements cut from finished parts, have been performed. Testing the laboratory samples made it possible to compare the obtained mechanical properties with the properties declared by PCM suppliers and to conduct a mor e correct assessment of the safety margins of the parts. To decide whether to install parts on the engine, fatigue and erosion tests of the structural elements cut from the finished parts were carried out. The final decision on the performance of the PCM parts was made after testing them as part of the upgraded LPC on the engine. The criterion for evaluating the erosion resistance of PCM parts has been introduced, which makes it possible to assess their performance during operation.

**Keywords:** composites; erosion; fatigue; finite element method; modal analysis

## 1. Introduction

The PCM usage in aviation and other industries has increased [1]. In aviation engines, the composites are already used in the production of fan blades, however this usage is possible for a limited number of companies. In Russian aviation engines, the composites can now be used for the manufacturing of non-rotating fan parts in the external air path.

The aviation engine LPC was upgraded in order to produce some PCM components (namely the flow separator (FS), the epoxy carbon fibers, and internal panels (IP)) from thermoplastic. The layout of the FS and IP in the LPC are shown in the Figure 1.

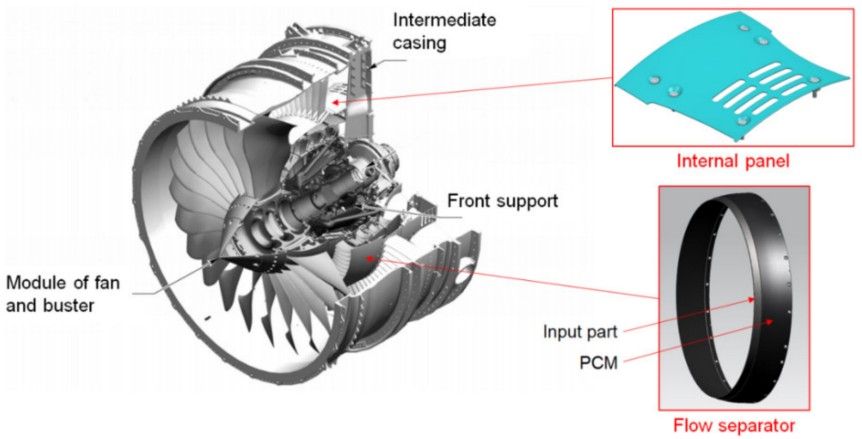

**Figure 1.** FS and inner panel in LPC.

There is a question about assessing the reliability and durability of such parts, since during operation they have different external loads:

- the static loading due to the difference between the pressure of the LPC on the internal surface and the pressure of the fan on the external surface,
- the dynamic loading due to the excitation of the parts by rotor harmonics,
- the entrance of sand and dust with the air stream, both as the plane takes off and lands, which leads to component erosion.

Static analysis is not currently a problem for anisotropic materials. Dynamics and erosion are more complex problems; however, there are many publications devoted to the practical assessment of the fatigue and erosion resistance of composites.

As usual, the studies of composite fatigue show the specimen tests, whereas real engine part tests are not presented. For example, A. J. Brunner [2] paid attention to the fact that, until a certain time, the approach to testing composites was based on the method of testing metals and alloys. The author emphasized that recently, some researchers have raised the issue of a detailed study of the fatigue fracture of composites. Test data is required for an understanding of the external sources of the scattering of fatigue, as well as to exploit the full potential of this material class in the design of the structures in which it is used.

N. S. Azikov and A. V. Zinin [3], using the finite element method, analyzed a structure made of a composite material for strength; the design model takes into account the degradation of the mechanical properties of the layers. The accumulated damage in the process of destruction was assessed and the maximum breaking load was determined.

T. D. Karimbaev and D. V. Matyukhin [4] showed that the load-bearing capacity of composite materials depends on the types of fatigue damage, both from each separately and from their combination with others. For this reason, at the moment there is no single law that can describe the fatigue behavior of a material in the development of composite structures. The authors suggested that the load-bearing capacity of the specimen is considered exhausted when the vibration frequency at the first bending form decreases to an acceptable value. After that, a correlation was established between the frequency, residual strength, and stiffness.

R. D. B. Sevenois and W. V. Paepegem [5] discussed the fatigue testing of polymer matrix composites, as well as various uniaxial and multiaxial test procedures, including the effects of grip and boundary conditions. The typical mechanisms of fatigue in polymer composites are presented, as well as the general control methods for detecting various types of fatigue damage.

Composite materials also find their application in the manufacturing of fan blades for aircraft engines [6–8].

Experiments with the injection of air and sand mixtures and the estimation of the erosion value, by measuring the erosion depth or specimen weight reduction, are the main test methods in most references. For example, T. D. Karimbaev and Yu. A. Nozhnitskii et al. [9] carried out an experimental study of the abrasive erosion of several composite materials and coatings based on polymers, carbon, metals, boron, organic fibers, and those reinforced with glass fiber. For comparison, several metal alloys that are widely used in mechanical engineering, ceramic, and elastic (based on rubber, polyethylene, adhesive films, etc.) coatings, and obtained by different methods, were tested.

V. S. Erasov and E. A. Kotova [10] analyzed the influence of erosion factors, as well as the physical and mechanical properties of the materials, on their erosion resistance. The authors also considered the possible mechanisms and types of erosional destruction and identified the main problems in the study of the erosion resistance of materials to the actions of solid particles.

V. A. Afanas'ev et al. [11] presented tools and methods for studying the properties of porous thermal insulation materials under the conditions of sudden pressure changes. The nature of the gas outflow is determined by the sample volume and the pressure drop, at which the sample is separated from the structure of the spacecraft element.

Bing Liu et al. [12] investigated the failure of solid particles in composites reinforced with polybenzoxazole (PBO) fibers in plain weave. Estimating the erosion rate of simple polymers and these composites at various angles of impact ($\alpha$ = 15–90°), the authors confirmed that the polymers and their composites are plastic. Based on the results, a mathematical model was created to predict the rate of the erosion of the plastic materials. The model was verified based on the comparison of the theoretical and measured data.

N. M. Barkoula et al. [13] reviewed the problem of the erosion of solids, in relation to the processes and modes during erosion, with an emphasis on polymer matrix composites.

The authors of [14] investigated the impact of the composite parts via air flow with the added sand in different concentrations. The attack angles and speeds of the air/sand mixture were chosen to model the erosion conditions for engine fan intake.

Most references do not consider the testing of real parts used in aviation engines. Therefore, the assessment of parts made from PCM, for use in aircraft engines, is important.

## 2. Theoretical Basis

The assessment of the parts made from PCM, for use in aircraft engines, requires a series of static and dynamic calculations and their subsequent verification during the experiments on test facilities and the engine.

The theory of the calculation of composite materials is described in sufficient detail, for example, in the ANSYS Theory Reference for the Mechanical APDL and Mechanical Applications.

However, the calculation of the parts made of composite materials requires the use of reliable material properties, which may depend on the production technology of the parts and significantly differ from the properties declared by the supplier of the source materials.

Besides some mechanical integrity properties, such as fatigue strength and erosion resistance, depend on part design and manufacture technology and cannot be calculated. Thus, the main task of this work was to develop a system of calculations and experiments that provide a confirmation of the possibility of using PCM for the manufacturing of aircraft engine parts and the installation of such parts on the engine.

This article describes the calculations and testing procedures for the FS and IP that could be performed prior to installation on the engine, which includes the following steps:

- calculation of the stress-strain state of the specimens, and carrying out static tests of the specimens for compliance with their mechanical properties, as declared by the manufacturer;
- static and dynamic calculations of the FS and IP;
- verification of the dynamic calculations by "ping" testing;
- fatigue testing of the whole FS and the structural elements cut from the FS;
- a creep test of the IP;
- examination of the cracked structural elements, using optical and electron microscopes;
- an erosion resistance assessment of the FS and IP.

These calculations and tests made it possible to obtain data on the mechanical properties of the PCM parts, as well as to estimate their safety margin.

## 3. Methodology

*3.1. Calculation and Static Tests of Specimens*

The calculation of the parts made of composite materials requires the use of reliable material properties, which may depend on the production technology of the parts and significantly differ from the properties declared by the supplier of the source materials. Therefore, the calculations and static testing of the specimens with thicknesses close to the thickness of the FS (and made from epoxy carbon using pre-preg and the technology of the FS manufacture), as well as the calculation and static testing of the specimens with thicknesses close to thickness of the internal panel (and made from thermoplastic carbon using pre-preg and the technology of internal panel manufacture) were proposed.

### 3.1.1. Material Properties

A pre-preg based on the epoxy carbon fiber Hexcel M56 [15] was chosen as the material for the FS, and a pre-preg based on the thermoplastic carbon fiber LPCL PEEK-4-40 [16] was chosen for the inner panels. The pre-preg properties declared by the manufacturer are shown in Table 1.

**Table 1.** Properties of materials.

| Properties | Epoxy Carbon | Thermoplastic Carbon |
|---|---|---|
| Elastic modulus E11, E22, MPa | 65,900 | 59,400 |
| Elastic modulus E33, MPa | 10,000 | 5714 |
| Poisson's ratio μ12 | 0.3 | 0.07 |
| Poisson's ratio μ23, μ13 | 0.3 | 0.15 |
| Shear modulus G12, MPa | 3500 | 4500 |
| Shear modulus G23, G13, MPa | 3200 | 4100 |
| Density, kg/m$^3$ | 1500 | 1530 |
| Glass transition temperature, °C | 182 | 148 |

### 3.1.2. Calculations of Specimens from Epoxy-Carbon Plastic and Thermoplastic

The calculations of the specimens were performed in order to estimate their stress-strain state and obtain the dependence of displacements on the load and stresses on deformations at different types of mechanical tests. Strength calculations were carried out by the finite element method in system ANSYS APDL, using the properties of the materials declared by the manufacturer.

The specimens were calculated for two models: 2D and 3D. Flat elements were used in 2D models, the thickness parameter of which corresponded with all the layers of the package, and the equivalent stiffness of the element was calculated based on the stiffness and stacking directions of the monolayers. This is the "engineering approach" to solving problems with layered composite materials, which, on the one hand, is very undemanding to computing resources and, on the other hand, allows accurate prediction of the stresses and deformations in each layer of the composite (and, therefore, predicts its destruction and describes its behavior under load quite well). The disadvantage of this method is the inability to estimate the stresses transverse to the fiber, for example, σ33.

In the 3D formulation, each of the monolayers of the composite material was defined as a separate three-dimensional layer, the properties and direction of which coincide with the properties of the monolayer. This method better describes the complex stress state of composite parts.

To compare the accuracy of both methods, in a number of cases, an analytical calculation was also carried out; although it does not take into account all the nuances of the behavior of the composite materials, the statement of the uniaxial stress state for a unidirectional package quite accurately describes both the stresses and deformations of the specimen.

The types of analyzed specimens and test loadings are shown in the Table 2.

### 3.1.3. Static Testing of Specimens

The specimens were made from the proposed materials using FS and IP technology to clarify their mechanical characteristics when tested in tension, compression, bending, and shear. The tests were carried out with an Instron 5882 electromechanical machine, as shown in Figure 2.

**Table 2.** Types of analyzed specimens and test loadings.

| No. | Specimen Material | Specimen's Dimensions | Specimen Type | Reinforcement Fiber Orientation | Load Type | Direction of Action of Forces towards the Base of the Reinforcing Fiber |
|-----|-------------------|----------------------|---------------|----------------------------------|-----------|--------------------------------------------------------------------------|
| 1 | Epoxy carbon | 250 × 25 × 2.56 | smooth | 0° | tension | along |
| 2 | | 175 × 25 × 2.56 | smooth | 90° | tension | across |
| 3 | | 150 × 12 × 2.56 | smooth | 0° | compression | along |
| 4 | | 140 × 15 × 2.56 | smooth | 90° | compression | across |
| 5 | | 60 × 12 × 2.56 | smooth | 0° | bend | across |
| 6 | | 50 × 10 × 5.14 | smooth | 0° | shear | across |
| 7 | | 300 × 36 × 2.56 | with hole | 0° | tension | along |
| 8 | | 150 × 100 × 5.14 | smooth | 0° | compression after impact | along |
| 9 | Thermoplastic carbon | 150 × 15 × 2.48 | smooth | 0° | tension | along |
| 10 | | 150 × 15 × 2.48 | smooth | 90° | tension | across |
| 11 | | 75 × 10 × 2.48 | smooth | 0° | compression | along |
| 12 | | 75 × 15 × 2.48 | smooth | 90° | compression | across |
| 13 | | 60 × 12 × 2.48 | smooth | 0° | bend | across |
| 14 | | 50 × 10 × 4.96 | smooth | 0° | shear | across |
| 15 | | 300 × 36 × 2.48 | with hole | 0° | tension | along |
| 16 | | 150 × 100 × 4.96 | smooth | 0° | compression after impact | along |

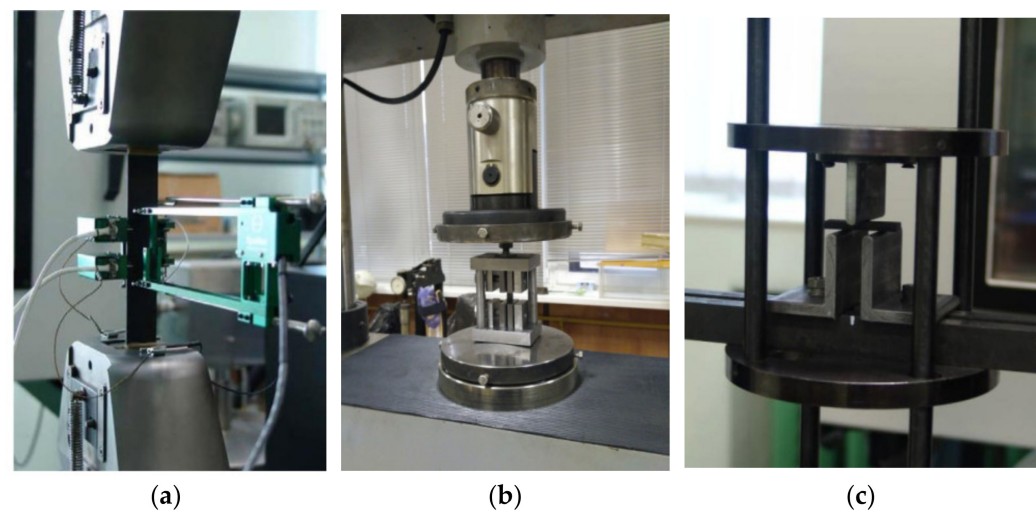

| **(a)** | **(b)** | **(c)** |

**Figure 2.** Testing machines for specifying the mechanical characteristics of specimens in tension (**a**), compression (**b**), and bending (**c**).

### 3.2. Calculations of FS and IP

The model and boundary conditions of the calculations of the FS and IP are described in [17] and [18], respectively.

The 3D dynamic calculation of the FS, as well as the 3D static and dynamic calculations of the IP, were carried out via the ANSYS package. The models of the composite FS and IP used SOLID186 elements.

The boundary conditions and the fixing of the separator and panel models were chosen in such a way as to simulate its real operating conditions on the engine, taking into account the tightening torque of the bolts. The temperature field and pressure acting on the panel during engine operation were applied to it.

### 3.3. Verification of Dynamic Calculations

To verify the calculations of the natural frequencies and vibration modes, a series of "ping" tests was carried out based on the impact modal analysis of the separator and panel at "free–free" and fixed conditions. Figure 3 shows the parts under testing at "free–free" conditions.

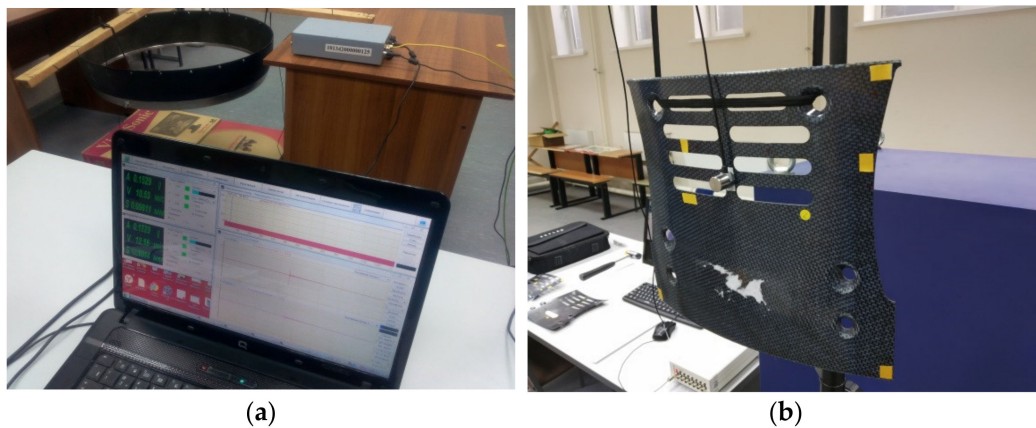

**Figure 3.** Determination of natural frequencies of the separator (**a**) and panel (**b**).

To determine the dynamic parameters at a given point of the investigated object, a piezoelectric accelerometer is installed (using glue or gel) and connected to the measuring system. Then, one hammer blow is applied at a predetermined point of the test object, normal to the surface of the panel.

The surface area of the impact should be large enough that the maximum force does not deform the cap of the hammer or the structure under test.

### 3.4. Fatigue Test of FS

#### 3.4.1. Fatigue Testing of the Whole FS

In order to reproduce fixation at the engine, the special adopter was designed for a fatigue test of the FS using a powerful vibrator [17].

#### 3.4.2. Fatigue Testing of Structural Elements

In order to carry out fatigue tests above the endurance limit of a material, it was suggested to carry out tests of the structural elements cut from FS, instead of the testing the whole separator. The test procedure is described in detail in [17].

The approach to testing with the minimum expenditure of resources that is outlined in this work makes it possible to determine the endurance limit of the composite materials from which the parts of the aircraft engines are made.

### 3.5. Creep Test of IP

The determination of the glass transition temperature for thermoplastic carbon specimens showed that this temperature is higher than the operating temperature by 12%.

The purpose of the creep test was to determine the residual deformation of the panel after exposure to the maximum load at the maximum operating temperature.

The test was carried out in a tooling using sandbags (shot), which are located between the inner surface of the panel, and the force loading system. The scheme to fix the panel in the tooling repeated the scheme to fix the panel to the engine (Figure 4).

With the help of the force loading system, the panel was unevenly loaded while its vertical displacement, relative to the fixture, was controlled at 10 preselected control points.

The layout of the control points was determined in accordance with the results of calculating the panel in the ANSYS finite element package under static loading, in a three-dimensional setting, and at the maximum engine operating mode.

The loaded panel was installed in the furnace and kept there at a constant temperature. After that, the panel was removed from the furnace, unloaded, cooled to room temperature, and measured at the control points. Then, the test cycle was repeated additional times.

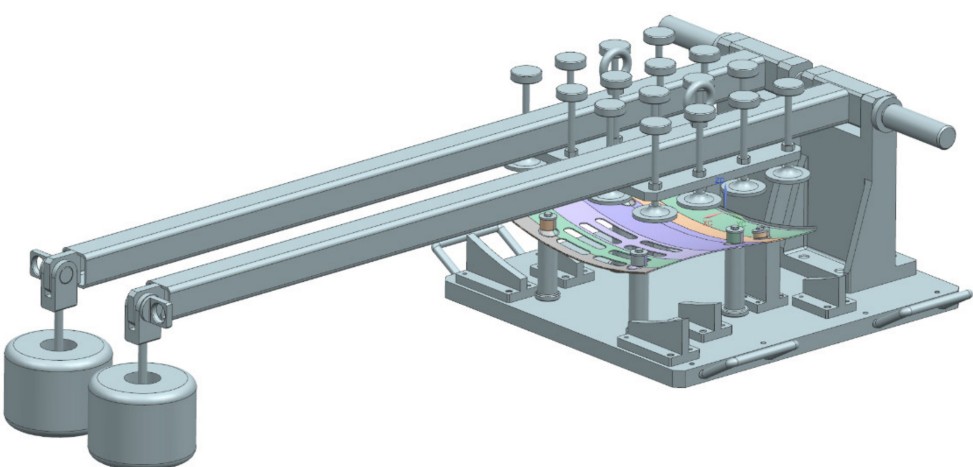

**Figure 4.** System for uneven panel loading during testing.

*3.6. Erosion Resistance Assessment*

The tests of the FS, with and without a protective coating and an internal panel for erosion resistance, were carried out in order to estimate the required level of resistance to abrasion in a stream of solid particles moving with a given speed and angle of attack.

Tests of the specimens were carried out on a gas-dynamic erosion installation, in which the impact on the test object is performed by forming a heterogeneous flow with the following parameters for the movement speed of the abrasive particles and their concentration in the flow: flow rate $200 \pm 20$ m/s and abrasive concentration $0.007 \pm 0.0005$ g/s

The erosive effect on the test object was carried out with a gas–powder mixture of air and ground quartz, which is prepared in the mixing chamber of the test bench.

The parameters of the test modes are set on the basis of simulating the process of throwing dust at the entrance to the engine fan, as well as the dynamics of further movements of individual dust fractions inside the engine.

Tests with a low-concentration of abrasion, as in flight, lead to long test durations. Therefore, the indirect method of erosion damage assessment was proposed for the internal panel and FS, detailed in [18,19].

This method is based on the measurement of residual thickness during engine operation. The evaluation of the natural frequency of the vibrations and maximum static stresses of the FS, with a different thickness of the composite part after erosion, have been chosen as the criterion for reliable operation. The number of specimens was not less than 5 for every group of the mechanical tests and was no less than 3 specimens for every group of other tests.

## 4. Results and Discussion

*4.1. Calculation and Static Tests of Specimens*

The test specimen consists of 12 layers with a thickness of 0.214 mm (with a total specimen thickness of 2.56 mm).

The calculation was performed in a non-linear setup, using a model consisting of 3780 elements and a corresponding 18,980 nodes (Figure 5a). The HEX20 element type is a three-dimensional, hexahedral, twenty-node quadratic element (SOLID186).

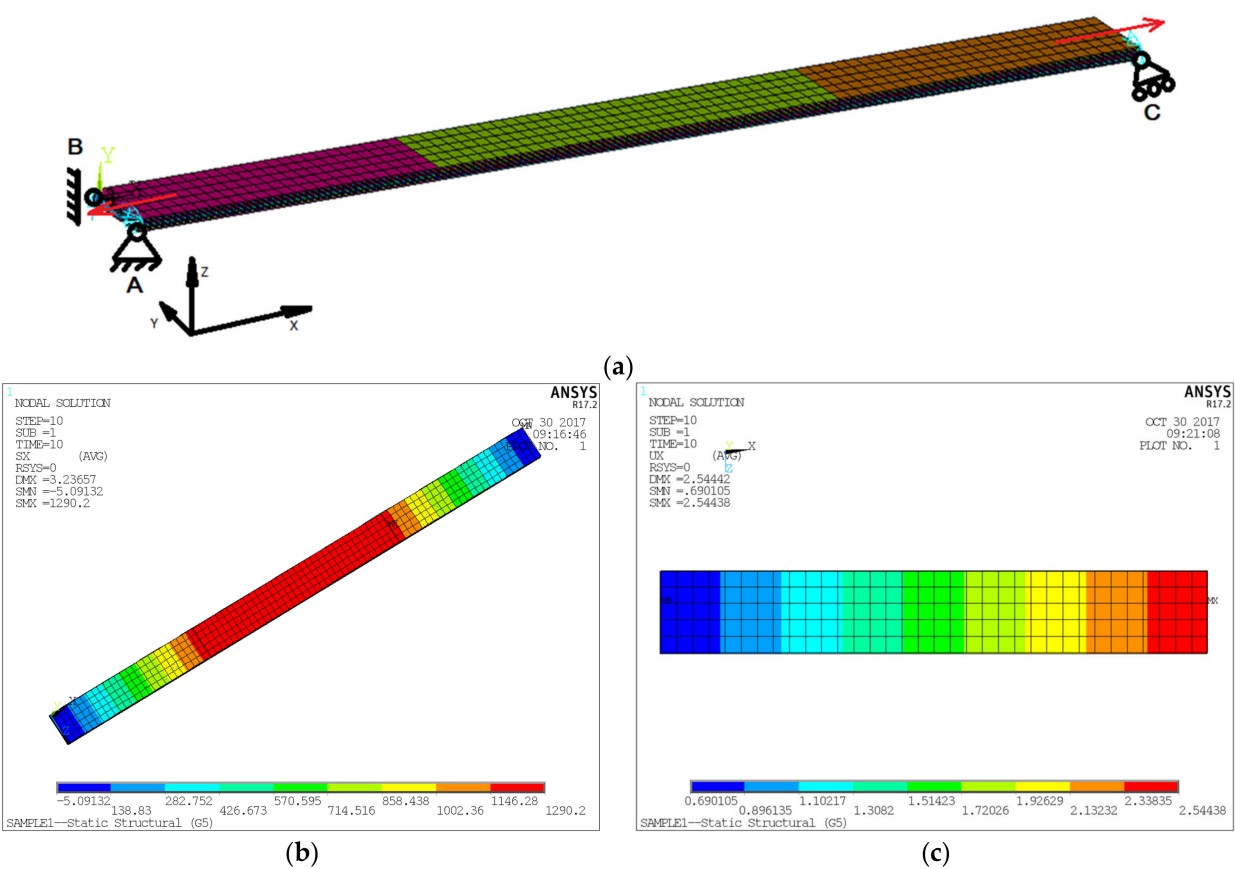

**Figure 5.** Example of boundary conditions (**a**), axial stresses (**b**) and axial displacements in the working area of the specimen (**c**) in the calculation of the specimen under tension and compression.

Since 3D models are sensitive to point sources of loads and fixing, a fixing scheme was implemented in this model; the model was fixed at three points: A (at the *X*, *Y*, and *Z* coordinates), B (at the *Y* coordinate), and at point C (at the *Z* and *Y* coordinates) (Figure 5a). This scheme allows us to fix the specimen in space, while not creating concentrators stresses. Since the area over which the load is applied will change during the solution under the influence of force, for the immutability of its magnitude, surface 8-node SURF154 elements were introduced, which takes this change into account and monitors the constancy of the applied force.

The calculation was carried out for the load value from 5 to 50 kN, in increments of 5 kN. Figure 5b,c shows the results obtained for the last loading step (50 kN).

A specimen for determining the destructive stress during the bending of an epoxy-carbon plastic with a reinforcing system was located in such a way that the base is parallel to the long side of the sample. Since 3D are models sensitive to point sources of loads and fixing, the following fixing scheme was implemented in this model: the model was fixed along two lines, corresponding to the contact surfaces of the support rollers A and B, at the *Y* and *Z* coordinates, as well as at one point in the middle of the model at the *X* coordinate (Figure 6a). With this scheme, the lower surface of the sample can deform between the supports without imposing additional stresses, which corresponds to the kinematic loading scheme implemented in the experiment.

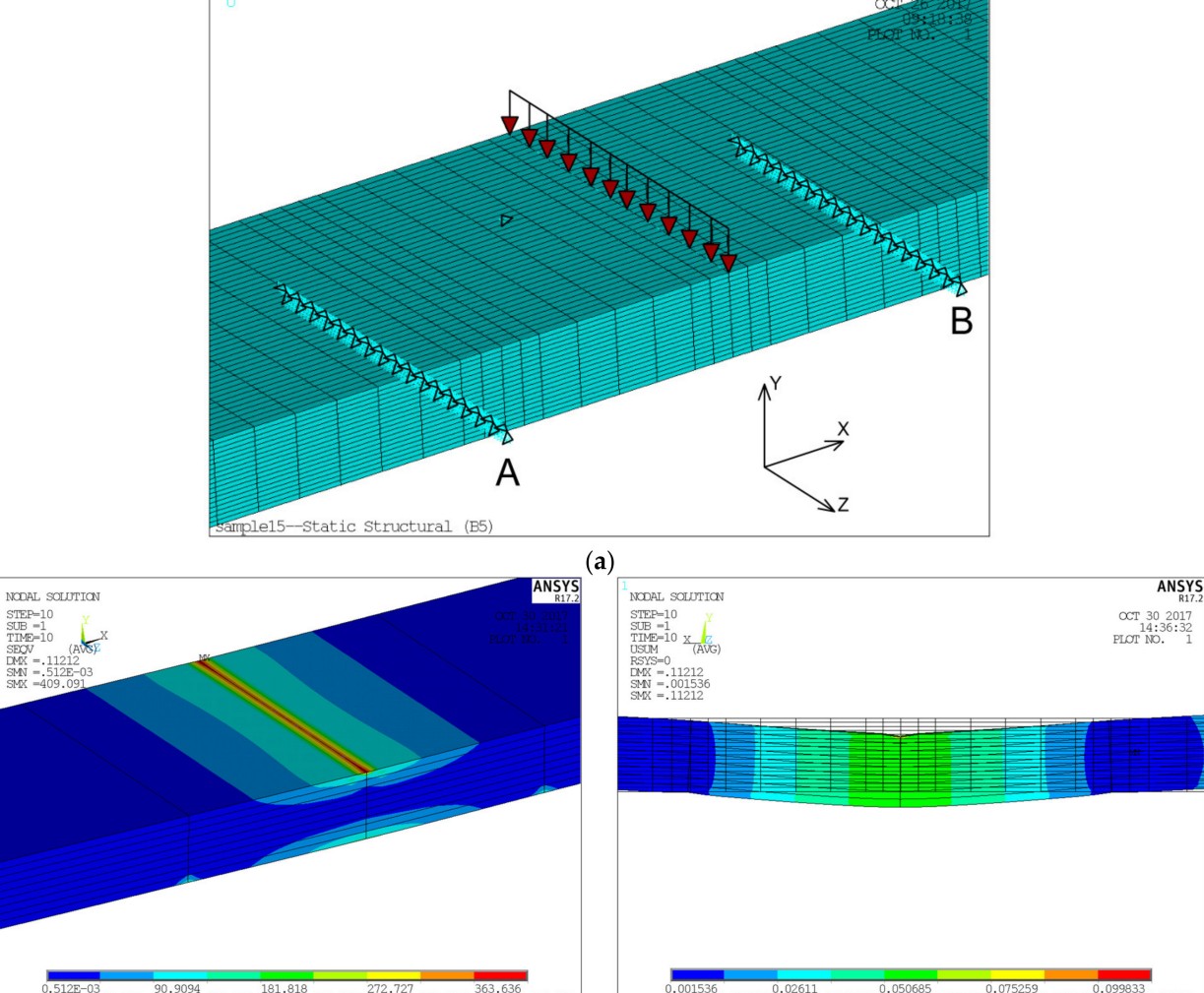

**Figure 6.** Example of the boundary condition (**a**), equivalent stress (**b**), and bending displacement (**c**) in the calculations of the specimen for interlayer bending.

The load was modeled as a force, applied along a line, passing through the center of the sample parallel to a short face (Figure 6a) in 10 steps of 50 N. The Figure 6b,c (below) shows the results obtained for the last loading step (500 N).

In the model of the specimen for shear calculation, the following fixation scheme was implemented: face A, B, and C are fixed in *Y* and *X*; face D is fixed in all directions (Figure 7a). In addition, the nodes belonging to the faces A, B, and C can only move in the Z direction together; that is, all nodes can only have the same movement. This condition follows from the method of fixing the real sample in the testing machine.

The load was modeled as a force, applied along Face A, as shown in Figure 7a. Figure 7b,c shows the results obtained for the last loading step (45 kN).

When calculating the specimen with a hole, the method of fixing and applying the load was used (similar to Figure 5). Figure 8 shows the results obtained for the last loading step (50 kN).

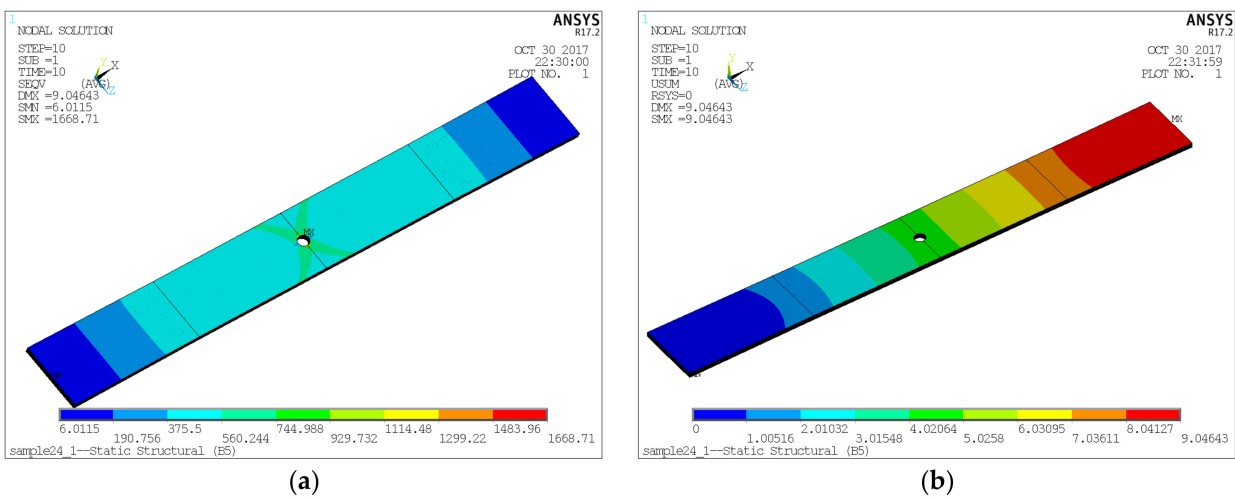

**Figure 7.** Example of boundary condition (**a**), equivalent stress (**b**), and shear displacement (**c**) in the calculations of the specimen for shear.

**Figure 8.** Example of equivalent stress (**a**) and displacement (**b**) calculations of the specimen with a hole under tension.

The calculations of the laboratory specimens of epoxy and thermoplastic carbons made it possible to choose the test modes of the specimens (expected ranges of loads and displacements for each type of test) and to show the expected criteria of failure.

After the calculations were carried out, some tests (for example, with a reinforcing angle of 45°) were found to be ineffective. Recommendations for the replacement and elimination of certain types of tests were taken into account when optimizing the design documentation for the laboratory specimens.

The static test results showed that a deviation of the mechanical elastic properties from the properties declared by the manufacturer is less than 3%, whereas the ultimate strength can be less by 30% for epoxy carbon and less by 41% for thermoplastic carbon.

*4.2. Calculations of FS and IP*

4.2.1. Dynamic Calculations of FS

The results of the FS calculations are described in [17]. The static calculation of the FS has not been performed, due to low level of expected static stress for the axisymmetric separator.

The revealed natural frequencies and modes of the FS oscillations are presented in Table 3.

**Table 3.** FS natural frequencies and vibration modes.

| Waveforms No. | Natural Vibration Frequencies, Hz |
|:---:|:---:|
| 1 | 308 |
| 2 | 333 |
| 3 | 417 |

For the separator, the first-and second-mode frequencies are close to the third rotor harmonic of the motor, and the third frequency can be close to the fourth rotor harmonic.

The FS high-cycle fatigue test should be carried out at the first, second, or third own frequencies, as determined by dynamic analysis.

4.2.2. Static and Dynamic Calculations of IP

The results of the static and dynamic calculations of IP are described in [18]. The results of the static calculation showed that the acting stresses in the panel make it possible to provide an acceptable margin of safety.

The differences between the calculated and test frequencies, when fixing on the engine, were no more than less than 3% for IP. The revealed natural frequencies and modes of the PS oscillations are presented in Table 4.

**Table 4.** Internal panel natural frequencies and vibration modes.

| Waveforms No. | Natural Vibration Frequencies, Hz |
|:---:|:---:|
| 1 | 518 |
| 2 | 559 |
| 3 | 580 |

For the internal panel, the first, second, and third own frequencies are close to the fifth rotor harmonic.

*4.3. Verification of Dynamic Calculations*

When exciting the process of the vibration of the structure with a shock and measuring the dynamic response, a spectral diagram of the damped oscillations is obtained, the peak values of which correspond to the experimental estimate of the frequencies of the natural oscillations of the test object (Figure 9).

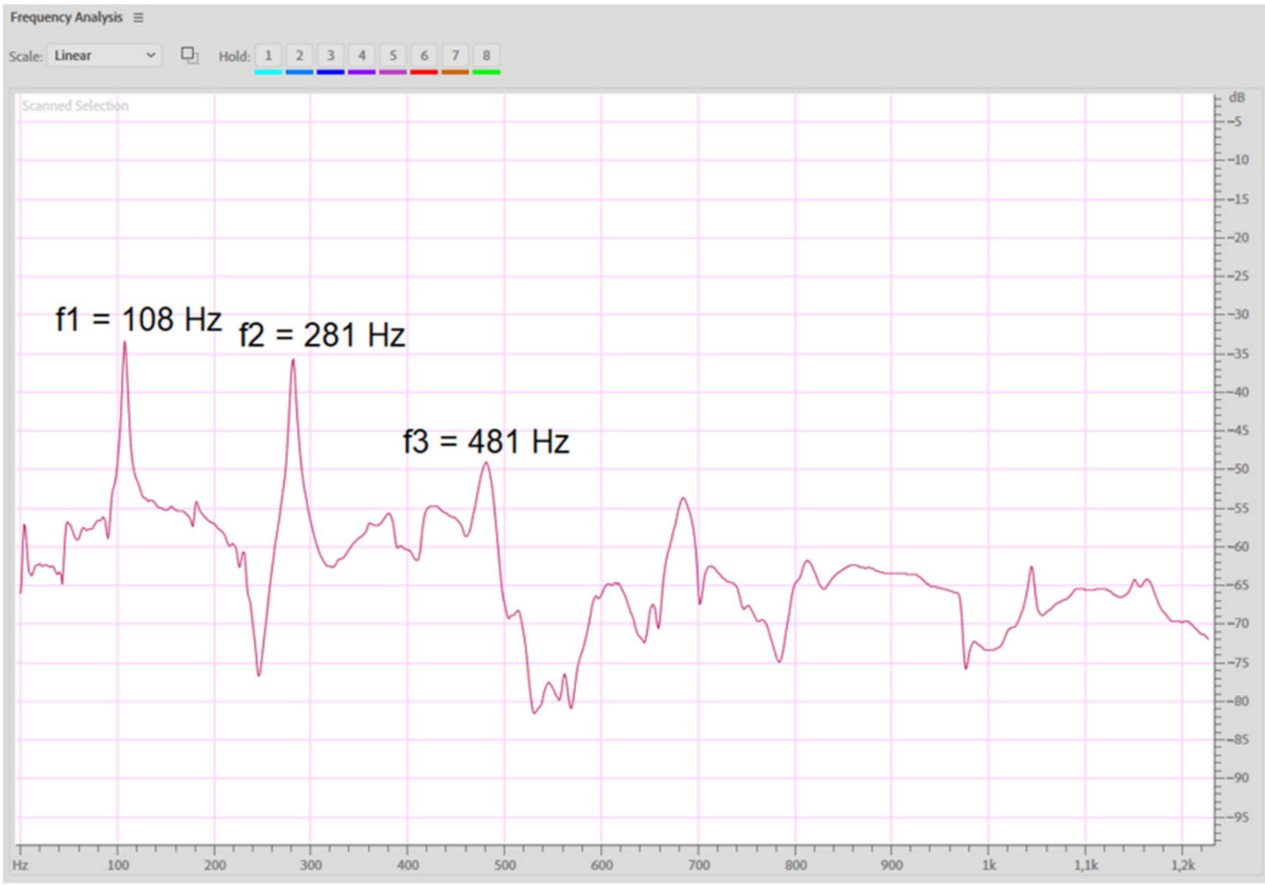

**Figure 9.** Oscillation diagram (spectrogram) of the test object (example).

The results of the dynamic calculations of the FS, under the conditions of "free–free", were compared with the results of the "ping" test and showed that the difference between the calculated and measured frequencies for FS is 20%, which can be explained by the scatter of material properties for different areas of details.

A similar comparison was performed for the IP and showed the difference between the calculated and measured frequencies for IP is less than 14%, which can be also explained by the scatter of the material properties for different areas of details.

### 4.4. Fatigue Test of FS

A stress stepwise increase test of whole-separator vibration, with a third-mode frequency, was used up to a basic $10^7$ cycle number. After vibration, on stresses 50 MPa during base cycles, the test was stopped due to a stand power limit, however fatigue damage has not been achieved.

Based on the results of the preliminary calculations, a series of fatigue tests were carried out on structural elements cut from the FS. The test procedure is described in [17].

According to the data obtained during the testing of the structural elements cut from the FS, a graph of the dependence of the cycle stress on the number of cycles was plotted (shown in the Figure 10).

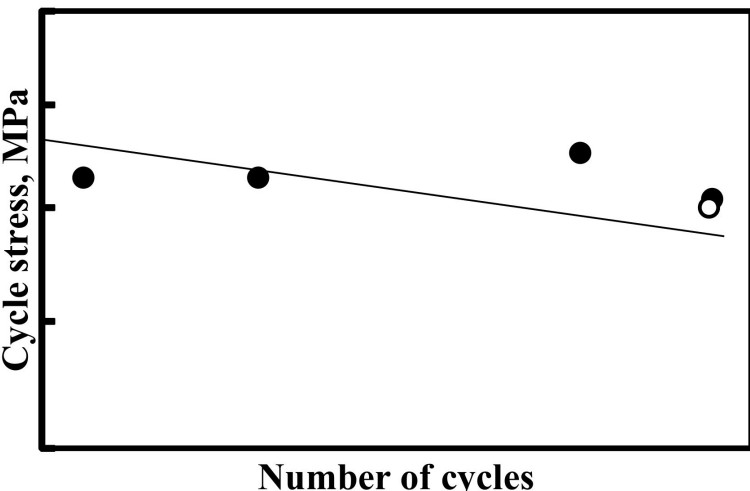

**Figure 10.** Fatigue curve of structural elements (●—has broken/suspected breakage; ○—did not break).

The elements with suspected breakage were examined under an optical and electron microscope in order to localize the areas with fatigue damage, after which the nature of said damage was assessed.

Three types of fatigue failure characteristics of a composite material were found: tearing off at the interfaces between the layers, chipping of the matrix material, and failure of the reinforcing fiber.

The ratio of the fatigue limit, found after testing the FS dynamic stress measured at engine, allowed the evaluation of the fatigue strength margin.

### 4.5. Creep Test of IP

The creep test results showed that certain displacements of the panel surface control points, after cyclic thermomechanical loading, do not exceed an unacceptable level. At the control point with the greatest deviation, the value of the displacements is 54% of the maximum allowable.

### 4.6. Erosion Resistance Assessment

Erosion resistance tests [14] of the samples were carried out at different abrasive mass flow rates and the angles of attack were measured between the trajectory of the abrasive particles and the surfaces of the samples. The obtained test results showed that the highest erosion resistance is achieved at small angles of attack (about 15 degrees) and the lowest is achieved at the direction of flow perpendicular to the sample surface.

Erosive wear is characterized by the loss of mass of the part; therefore, its natural frequency will also change. In this regard, the dynamic and static analyses were performed on the FS and the internal panel, with different thicknesses of the composite parts exposed to erosion (the results of which are detailed in [18,19]).

The calculation results showed that the thickness of the FS should be reduced by no more than 0.8 mm, above which the natural frequency of the FS approaches the exciting harmonic of the motor (315 Hz).

For a 0.9 mm thick IP, the calculated natural frequency of the first vibration mode is 442 Hz. Due to the proximity of this frequency to the rotor harmonics, there is a danger of its critical operation. However, static stresses, in comparison with stresses, at nominal panel thickness increased by 40% and did not exceed the permissible material stress level.

The results of the computational studies showed that a decrease in the thickness of a part is allowed for up to 1 mm, provided that sufficiently accurate measurements are obtained during the field control of an aircraft engine.

## 5. Conclusions

This article considers the methodology for assessing the impact of operational factors on the operation of aircraft engine parts made of composite materials. Using the example of a composite flow separator and a composite internal panel, a chain of measurements is demonstrated, which allows us to draw conclusions about the reliability of PCM products. The combination of the experimental determination of the properties of the materials and the finished product (with the preservation of the technological features of the parts), as well as the methods of mathematical modeling and experimental evaluation of the resource allows us to evaluate the strength of the PCM products with high reliability, which will help in the further design of such parts.

In the cycle of the work carried out, the following results were obtained:

- testing the laboratory samples allowed us to compare the obtained mechanical properties with the properties declared by PCM suppliers and to conduct a more correct assessment of the safety margin of parts;
- the stress-strain state of the flow separator and the inner panel made of PCM was calculated, taking erosive wear into account, and showed a decrease in the safety margin from 2.56 to 1.85 with a decrease in thickness of 33%;
- the natural oscillation frequencies and dynamic voltages of the flow separator and the internal panel, made of PCM, are determined. The confirmation of the results of the dynamic calculations was performed using the "ping" test. The critical thicknesses of the FS and IP are determined, at which the natural frequencies of the parts approach the rotary harmonics of an aircraft engine;
- an assessment of the residual deformations of the inner panel, after simultaneous exposure to the maximum operating load and temperature, was carried out.

To decide whether to install parts on the aircraft engine, fatigue and erosion tests of the structural elements cut from the finished parts were carried out, as well as testing the parts for vibration resistance, cyclic strength, and creep.

The final decision on the performance of the PCM parts was made after testing them as part of the upgraded LPC on the aircraft engine.

The criterion for evaluating the erosion resistance of PCM parts has been introduced, which makes it possible to assess their safety during operation.

**Author Contributions:** Conceptualization, Y.R. and A.A.; methodology, A.S. and T.E.; software, A.A. and A.S.; validation, Y.R., A.A., A.S., and T.E.; formal analysis, A.A.; investigation, T.E.; resources, Y.R.; data curation, A.S. and T.E.; writing—original draft preparation, A.A. and A.S.; writing—review and editing, Y.R. and T.E.; visualization, A.S.; supervision, A.A. and T.E.; project administration, T.E.; funding acquisition, Y.R. All authors have read and agreed to the published version of the manuscript.

**Funding:** This research received no external funding.

**Institutional Review Board Statement:** Not applicable.

**Informed Consent Statement:** Not applicable.

**Data Availability Statement:** Not applicable.

**Conflicts of Interest:** The authors declare no conflict of interest. The funders had no role in the design of the study (including the collection, analyses, or interpretation of the data), writing of the manuscript, or decision to publish the results.

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
