# Peer review of "Application of Composite Materials in an Upgraded Engine Low-Pressure Compressor for a Regional Passenger Aircraft"

_inventions, doi:10.3390/inventions6030054_

Round 1

Reviewer 1 Report

This paper presents the well-composed analysis of successful evaluation of highly robust components produced from polymer composite materials for the low-pressure compressor.

Such a topic is fascinating in the application’s view because such practical work does give an instructive result that can be implemented very quickly in aircraft manufacturing.

 Review of Literature:

The author cited a properly composed review of the literature, and a lot of appropriate references were used in the introduction section. These statements contributed generously to the overall understanding of the subject and the reasoning for establishing the problem statement, thorough analysis of polymer composite materials usage in aviation and other industries.

Methodology:

The general methodology and description are well prepared. The action plan is clearly defined and well thought out except for the unpredictable time to iteratively improve individual modeling and measurement methods. The investigation methods and techniques used to gather the data for this article have been adequately selected, justified, and finally clearly discussed and concluded. However, the measurement techniques were explained, but the reliability coefficients of all possible tests were not given, and no discussion of the statistical methods was presented in this particular section.

Findings and conclusions

The findings were well organized, sectioned, and reported objectively. The charts were well presented, but the difficulty of the statistical tests employed would not be stand-alone to the average reader.

A solid point of the article is the high feasibility of using the reported issues’ results. The presented concept of applying the solution resulting from the conducted experiment is feasible and valuable for science and industry.

Language:

The English is almost perfect; I have not found only some tiny mistakes in language that need corrections:

Line 42: It appears that you are missing a comma after the introductory phrase As usual. Consider adding a comma.

Line 44: It seems that you are missing a comma after “time”. Consider adding a comma.

Line 50: The indefinite article, a, may be redundant when used with the uncountable noun material in your sentence. Consider removing it from “a composite”.

Line 52: It appears that you are missing a comma before the coordinating conjunction and in a compound sentence. Consider adding a comma.

Line 67: The noun phrase injection seems to be missing a determiner before it. Consider adding an article.

Line 92: The noun phrase impact seems to be missing a determiner before it. Consider adding an article

Line 185: The verb was does not seem to agree with the subject. Consider changing the verb form (was => were)

Author Response

Point 1: The general methodology and description are well prepared. The action plan is clearly defined and well thought out except for the unpredictable time to iteratively improve individual modeling and measurement methods.

Response 1: We didn’t understand this comment.

Point 2: The investigation methods and techniques used to gather the data for this article have been adequately selected, justified, and finally clearly discussed and concluded. However, the measurement techniques were explained, but the reliability coefficients of all possible tests were not given, and no discussion of the statistical methods was presented in this particular section.

Response 2: We didn’t carry out detailed statistical investigation taking into account limited number of ready parts however the number of specimens was not less than 6 for every group of mechanical test and no less than 3 specimens for every group of other tests.

Point 3: Line 42: It appears that you are missing a comma after the introductory phrase As usual. Consider adding a comma.

Response 3: Comma was added.

Point 4: Line 44: It seems that you are missing a comma after “time”. Consider adding a comma.

Response 4: Comma was added.

Point 5: Line 50: The indefinite article, a, may be redundant when used with the uncountable noun material in your sentence. Consider removing it from “a composite”.

Response 5: The indefinite article, a, was removed.

Point 6: Line 52: It appears that you are missing a comma before the coordinating conjunction and in a compound sentence. Consider adding a comma.

Response 6: Comma was added.

Point 7: Line 67: The noun phrase injection seems to be missing a determiner before it. Consider adding an article.

Response 7: In our opinion, a determiner is not needed here.

Point 8: Line 92: The noun phrase impact seems to be missing a determiner before it. Consider adding an article

Response 8: In our opinion, a determiner is not needed here.

Point 9: Line 185: The verb was does not seem to agree with the subject. Consider changing the verb form (was => were) This sentence already uses the verb "were".

Response 9: This sentence already uses the verb "were".

Reviewer 2 Report

First, I'd like to thank you for allowing me to review this article. Unfortunately, I see a lot of inaccuracies and inaccuracies in it.
There is a glaring mistake in the first paragraph of "Introduction". Composites are already used in the production of fan blades (and even complete compressor rotors). The first paragraph suggests that only non-rotating parts are made of composites. The subject of the publication itself seems interesting. Unfortunately, the introduction and review of the literature do not deal with the working conditions and loads that affect the components of aircraft engines at all. The authors only write about composites, ignoring the research object. Also, no reference was made to the aircraft engine in the conclusions.
Boundary conditions related to temperature and pressure were mentioned in the paper, but these values ​​were not presented (and the source of this information was not referred to).
The authors showed numerical analysis results, but the results are illegible (to scale) and incorrectly described (what stresses: reduced Von-Mises or Tresci-Coulomb, shear, principal?). In addition, no information was provided on the failure criteria for composite structures, which is essential in numerical tests (Tsai-Wu, Tsai-Hill, Hashin, Puck, LaRC)?
A lot of generalities were used in the description of the fatigue tests, without referring to specific results.
The quality of some of the graphics in the work is also unacceptable, e.g. figure 4 or 9.
In discussing the results of modal tests, no reference was made to the operating temperature (and this may have a significant impact on the frequency of resonant vibrations of the tested objects

There were other questions during the review:
- why the strength analysis of the entire separator or panel was not carried out?
- Is chapter 4.2 not too close to publication [12]?

There are individual errors in the edition of the publication, but they do not affect the reception of the work.

Author Response

Point 1: There is a glaring mistake in the first paragraph of "Introduction". Composites are already used in the production of fan blades (and even complete compressor rotors). The first paragraph suggests that only non-rotating parts are made of composites. The subject of the publication itself seems interesting. Unfortunately, the introduction and review of the literature do not deal with the working conditions and loads that affect the components of aircraft engines at all. The authors only write about composites, ignoring the research object. Also, no reference was made to the aircraft engine in the conclusions.

Response 1: The first paragraph of "Introduction" is revised. We don’t agree with comment the introduction and review of the literature do not deal with the working conditions and loads that affect the components of aircraft engines at all. The first paragraph on the page 2 lists external loads. Reference was made to the aircraft engine in the conclusions.

Point 2: Boundary conditions related to temperature and pressure were mentioned in the paper, but these values were not presented (and the source of this information was not referred to).

Response 2: Volume of the article don’t allow to give distributions of temperatures and gas loads on the external and internal surfaces of flow separator and internal panel. These distributions are calculated during usual aerodynamic and heat transfer calculations that are not subjects of this article.

Point 3: The authors showed numerical analysis results, but the results are illegible (to scale) and incorrectly described (what stresses: reduced Von-Mises or Tresci-Coulomb, shear, principal?). In addition, no information was provided on the failure criteria for composite structures, which is essential in numerical tests (Tsai-Wu, Tsai-Hill, Hashin, Puck, LaRC)?

Response 3: The descriptions of stresses and displacements are added in Figures 5-8. The scales have been deleted in accordance with security requirements. The failure criteria have not been estimated because strength margins have been evaluated as Ratio of the fatigue limit found after test to FS dynamic stress measured at engine test allowed (see last paragraph of chapter 4.4).

Point 4: A lot of generalities were used in the description of the fatigue tests, without referring to specific results.

Response 4: The specific results have been deleted in accordance with security requirements.

Point 5: The quality of some of the graphics in the work is also unacceptable, e.g. figure 4 or 9.

Response 5: Figures 4 and 9 are revised.

Point 6: In discussing the results of modal tests, no reference was made to the operating temperature (and this may have a significant impact on the frequency of resonant vibrations of the tested objects).

Response 6: Comparison of frequencies calculated at operating and room temperatures showed that difference is less than 1.4%.

Point 7: why the strength analysis of the entire separator or panel was not carried out?

Response 7: The strength analysis of the entire separator and panel was carried out and shown in the tables 3 and 4 and presented more detail in references [12] and [13]

Point 8: Is chapter 4.2 not too close to publication [12]?

Response 8: Chapter 4.2 is extract of publication [12].

Reviewer 3 Report

This manuscript describes the computational and experimental work carried out in order to evaluate the robustness and durability of components produced of polymer composite materials (PCM) as a part of the modernization of the low-pressure compressor (LPC) of the engine for the regional aircraft. It should be noted that the temperature of the air is increased at the outlet of the compressor (the air is compressed). The authors haven’t considered the temperature effects on the strength of the composite materials. The composite strength is highly influenced by the temperature. For example: the creep rupture stress of the composite decreases considerably with increasing the temperature. In my opinion the authors should solve the heat conduction equation of the composite material. The results produced in the finite element method aren’t clear. The quality of these figures needs to be improved. The authors should replace them. The authors should specify the boundary conditions, model assumptions and element size. The authors should perform major revisions on this manuscript before it can be considered for publication in Inventions journal.

Comments and Suggestions for Authors

1) (Abstract - Lines no. 8) the words “in order” should be added before the word “to”.

2) (Introduction section) some of the abbreviations used inside the paper should be defined and explained in the beginning of the abstract and the introduction.

3) (Section 3.1.1) the mechanical properties of the Carbon epoxy and the thermoplastic carbon shown presented in table 1 should be referenced.

4)  The glass transition temperature of the epoxy resin should be described in table 1. The glass temperature value is the temperature at which amorphous polymers change from hard to soft.

5)  The heading of sections 4 and 4.1 should move to the following page.

6) (Section 4.1 – figures 7 & 8) these figures are blurred. The quality of these figures needs to be improved. The authors should replace them. The authors should specify the boundary conditions, model assumptions and element size. What are the values of the displacements and stresses shown in these figures?

7) (Section 4.5) this section should be extended. The creep rupture stress of the composite decreases considerably with increasing the temperature. In my opinion the authors should solve the heat conduction equation of the composite material. The rupture time of the composite parts should be described.

8) (Section 4.6) the authors should present the erosion and wear values of the composite. I have looked at your cited papers (References no. 13 and 14). This information is missing. It is important factor in composite part design. The following paper should be cited:

Alqallaf, J.; Ali, N.; Teixeira, J.A.; Addali, A. Solid Particle Erosion Behaviour and Protective Coatings for Gas Turbine Compressor Blades—A Review. Processes 2020, 8, 984. https://doi.org/10.3390/pr8080984.

9) The conclusion section should be extended.

10) (Reference section) the authors should mention the DOI of the papers (See the following reference example):

Author 1, A.B.; Author 2, C.D. Title of the article. Abbreviated Journal Name Year, Volume, page range, DOI.

Author Response

Point 1: This manuscript describes the computational and experimental work carried out in order to evaluate the robustness and durability of components produced of polymer composite materials (PCM) as a part of the modernization of the low-pressure compressor (LPC) of the engine for the regional aircraft. It should be noted that the temperature of the air is increased at the outlet of the compressor (the air is compressed). The authors haven’t considered the temperature effects on the strength of the composite materials. The composite strength is highly influenced by the temperature. For example: the creep rupture stress of the composite decreases considerably with increasing the temperature. In my opinion the authors should solve the heat conduction equation of the composite material.

Response 1: Volume of the article don’t allow to give distributions of temperatures on the external and internal surfaces of flow separator and internal panel. These distributions were calculated during usual aerodynamic and heat transfer calculations that are not subjects of this article.

Point 2: The results produced in the finite element method aren’t clear. The quality of these figures needs to be improved. The authors should replace them. The authors should specify the boundary conditions, model assumptions and element size. The authors should perform major revisions on this manuscript before it can be considered for publication in Inventions journal.

Response 2: As it can be seen in table 2, 16 types of specimens with different dimensions and scheme of loading have been calculated and tested. Detail specifications of boundary conditions, model assumptions and element sizes could be problem due to limitation of article volume.

Point 3: 1) (Abstract - Lines no. 8) the words “in order” should be added before the word “to”.

Response 3: The words “in order” was added.

Point 4: 2) (Introduction section) some of the abbreviations used inside the paper should be defined and explained in the beginning of the abstract and the introduction.

Response 4: All of abbreviations were allready defined and explained in the beginning of manuscript.

Point 5: 3) (Section 3.1.1) the mechanical properties of the Carbon epoxy and the thermoplastic carbon shown presented in table 1 should be referenced.

Response 5: References include in article.

Point 6: 4) The glass transition temperature of the epoxy resin should be described in table 1. The glass temperature value is the temperature at which amorphous polymers change from hard to soft.

Response 6: The glass transition temperature of the epoxy resin is added to table 1 and in the beginning of paragraph 3.5.

Point 7: The heading of sections 4 and 4.1 should move to the following page.

Response 7: The heading of sections 4 and 4.1 were moved to the following page.

Point 8: 6) (Section 4.1 – figures 7 & 8) these figures are blurred. The quality of these figures needs to be improved. The authors should replace them. The authors should specify the boundary conditions, model assumptions and element size. What are the values of the displacements and stresses shown in these figures?

Response 8: Blurred figures are replaced. Detail specifications of boundary conditions, model assumptions and element sizes could be problem due to limitation of article volume (see answer to comment 2).

Point 9: 7) (Section 4.5) this section should be extended. The creep rupture stress of the composite decreases considerably with increasing the temperature. In my opinion the authors should solve the heat conduction equation of the composite material. The rupture time of the composite parts should be described.

Response 9: Temperature distributions were calculated during usual aerodynamic and heat transfer calculations that are not subjects of this article. Taking into account that (1) the glass transition temperature determined during special test is higher than operating temperature by 12% and (2) no reliable creep data for calculation of creep rupture time it was decided to carry out special cyclic creep test of whole panels instead of unreliable calculations.

Point 10: 8) (Section 4.6) the authors should present the erosion and wear values of the composite. I have looked at your cited papers (References no. 13 and 14). This information is missing. It is important factor in composite part design. The following paper should be cited: Alqallaf, J.; Ali, N.; Teixeira, J.A.; Addali, A. Solid Particle Erosion Behaviour and Protective Coatings for Gas Turbine Compressor Blades—A Review. Processes 2020, 8, 984. https://doi.org/10.3390/pr8080984.

Response 10: The results and methods of carrying out work on the research of erosion are presented in the work published [11]. A reference to the work has been added to clause 4.6.

Point 11: 9) The conclusion section should be extended.

Response 11: The conclusion section has been extended.

Point 12: 10) (Reference section) the authors should mention the DOI of the papers (See the following reference example):
Author 1, A.B.; Author 2, C.D. Title of the article. Abbreviated Journal Name Year, Volume, page range, DOI.

Response 12: DOI has been added for articles that have it.

Round 2

Reviewer 2 Report

Thanks for the answers. I accept the article after the changes.

Author Response

Finite element analysis should include also boundary conditions and initial conditions (if it is time dependent problem).

Done for figures 5-8.

Finite element analysis of the flow separator and internal panel more describe in references [12] and [13].

The magnitude of the stresses and displacement should be shown in figures 5-8.

Done.

Reviewer 3 Report

Thank you. Regarding Response 2:  Finite element analysis should include also boundary conditions and initial conditions (if it is time dependent problem). The magnitude of the stresses and displacement should be shown in figures 5-8. 

Author Response

(The authors gave the same response as above.)
